# Essential role of FBXL5-mediated cellular iron homeostasis in maintenance of hematopoietic stem cells

Yoshiharu Muto[1], Masaaki Nishiyama[1,*], Akihiro Nita[1,*], Toshiro Moroishi[1] & Keiichi I. Nakayama[1]

Hematopoietic stem cells (HSCs) are maintained in a hypoxic niche to limit oxidative stress. Although iron elicits oxidative stress, the importance of iron homeostasis in HSCs has been unknown. Here we show that iron regulation by the F-box protein FBXL5 is required for HSC self-renewal. Conditional deletion of *Fbxl5* in mouse HSCs results in cellular iron overload and a reduced cell number. Bone marrow transplantation reveals that FBXL5-deficient HSCs are unable to reconstitute the hematopoietic system of irradiated recipients as a result of stem cell exhaustion. Transcriptomic analysis shows abnormal activation of oxidative stress responses and the cell cycle in FBXL5-deficient mouse HSCs as well as downregulation of *FBXL5* expression in HSCs of patients with myelodysplastic syndrome. Suppression of iron regulatory protein 2 (IRP2) accumulation in FBXL5-deficient mouse HSCs restores stem cell function, implicating IRP2 as a potential therapeutic target for human hematopoietic diseases associated with FBXL5 downregulation.

[1] Department of Molecular and Cellular Biology, Medical Institute of Bioregulation, Kyushu University, 3-1-1 Maidashi, Higashi-ku, Fukuoka, Fukuoka 812-8582, Japan. * These authors contributed equally to this work. Correspondence and requests for materials should be addressed to M.N. (email: nishiyam@bioreg.kyushu-u.ac.jp) or to K.I.N. (email: nakayak1@bioreg.kyushu-u.ac.jp).

Hematopoietic stem cells (HSCs) are the most undifferentiated cells in the mammalian hematopoietic system, which they maintain throughout life. At steady state, HSCs are quiescent and reside in their hypoxic niche. They expend energy mostly via anaerobic metabolism by maintaining a high rate of glycolysis. These characteristics promote HSC maintenance by limiting the production of reactive oxygen species (ROS)[1], to which HSCs are highly vulnerable compared with other hematopoietic cells[2]. Homeostasis of cellular iron, which is a major elicitor of ROS production, is thus likely to be strictly regulated in HSCs in order for them to maintain their stemness.

Iron is essential for fundamental metabolic processes in cells and organisms, and it is incorporated into many proteins in the form of cofactors such as heme and iron–sulfur clusters. Iron also readily participates in the Fenton reaction, however, resulting in uncontrolled production of the hydroxyl radical, which is the most harmful of ROS in vivo and damages lipid membranes, proteins and DNA. It is therefore important that cellular iron levels are subject to regulation[3]. We previously showed that iron homeostasis in vivo is regulated predominantly by F-box and leucine-rich repeat protein 5 (FBXL5) and iron regulatory protein 2 (IRP2)[4]. IRP2 functions as an RNA binding protein to regulate the translation and stability of mRNAs that encode proteins required for cellular iron homeostasis. IRP2 thereby increases the size of the available iron pool under iron-limiting conditions. In contrast, under iron-replete conditions, FBXL5, which is the substrate recognition component of the SCF$^{FBXL5}$ E3 ubiquitin ligase, mediates ubiquitylation and degradation of IRP2. Whereas FBXL5 is unstable under iron-deficient conditions, direct binding of iron to its hemerythrin domain stabilizes the protein, with this iron-sensing ability allowing FBXL5 to control the abundance of IRP2 in an iron-dependent manner[5,6]. Disruption of the Fbxl5 gene in mice results in the failure of cells to sense increased cellular iron availability, which leads to constitutive accumulation of IRP2 and misexpression of its target genes. FBXL5-null mice die during embryogenesis as a result of overwhelming oxidative stress, indicating the vital role of FBXL5 in cellular iron homeostasis during embryogenesis[4].

A substantial proportion of iron in the adult body is present in the liver and hematopoietic system. Excess iron in the liver is clinically important given that cirrhosis and hepatocellular carcinoma often develop in individuals with systemic iron-overload disorders[7]. Conditional FBXL5 deficiency in mouse liver was found to result in iron accumulation and mitochondrial dysfunction in hepatocytes, leading to the development of steatohepatitis[4]. In contrast, hematopoiesis is sensitive to iron deficiency, with an insufficiency of available iron in the body being readily reflected as iron-deficiency anaemia[8].

Iron overload in the haematopoietic system is also clinically important, however. Systemic iron overload is thus frequently associated with hematologic diseases such as myelodysplastic syndrome (MDS), a clonal HSC disorder characterized by hematopoietic failure as a result of ineffective hematopoiesis[9–11]. Such iron overload is a consequence of the inevitability of frequent blood transfusions and suppression of hepcidin production as a result of ineffective erythropoiesis[12]. Clinical evidence suggests that systemic iron overload has a suppressive effect on hematopoiesis in individuals with MDS or aplastic anaemia, and that iron-chelation therapy often improves this situation[13–15]. These observations thus imply that hematopoietic failure promotes systemic iron overload, which in turn exacerbates hematopoietic failure, with the two conditions forming a vicious cycle. Oxidative stress was found to be increased in bone marrow (BM) cells of patients with iron overload, and the impaired hematopoietic function of these individuals was partially rescued by treatment with an antioxidant or iron chelator,

suggestive of the initial presence of ROS-induced cellular injury[16]. However, the molecular mechanisms underlying hematopoietic suppression by systemic iron overload in patients as well as the cell-autonomous effect of cellular iron overload on HSC stemness have remained largely unknown.

Here, we show that cellular iron homeostasis governed by the FBXL5–IRP2 axis is integral to the maintenance of HSCs. Ablation of FBXL5 specifically in the hematopoietic system of mice resulted in cellular iron overload in HSCs and impaired their ability to repopulate BM. FBXL5 was also found to be indispensable for the resistance of HSCs to stress induced by myelotoxic agents. FBXL5-deficient HSCs manifested oxidative stress, increased exit from quiescence and eventual exhaustion. Of note, FBXL5 expression was shown to be downregulated in HSCs of some MDS patients, suggesting that disruption of cellular iron homeostasis contributes to hematopoietic failure in such individuals by compromising HSC function. Suppression of IRP2 activity in FBXL5-deficient HSCs restored stem cell function, implicating IRP2 as a potential novel therapeutic target in stem cell diseases such as MDS that are associated with cellular iron overload.

## Results

**Cellular iron homeostasis is essential for HSC maintenance.** We first examined the expression of Fbxl5 in various hematopoietic cell lineages of wild-type mice by reverse transcription (RT) and real-time polymerase chain reaction (rtPCR) analysis. FBXL5 mRNA was detected in many cell lineages including HSCs (Supplementary Fig. 1a). Among differentiated cells, FBXL5 mRNA was most abundant in myeloid (Gr1$^+$Mac1$^+$) cells and least abundant in the erythroid (Ter119$^+$) lineage. These findings are largely consistent with the results of previous microarray[17] (Supplementary Fig. 1b) and RNA-sequencing[18,19] (Supplementary Fig. 1c) analyses.

To explore the role of cellular iron homeostasis in the maintenance and function of HSCs, we generated mice in which deletion of Fbxl5 is inducible in the hematopoietic system. Crossing of Fbxl5$^{F/F}$ mice (which harbour floxed alleles of Fbxl5) with Mx1-Cre transgenic mice (which express Cre recombinase under the control of the Mx1 gene promoter) followed by intraperitoneal injection of the resulting offspring (Mx1-Cre/Fbxl5$^{F/F}$ mice) with poly(I)–poly(C) [poly(I:C)] gives rise to the Fbxl5$^{\Delta/\Delta}$ genotype in HSCs. Mx1-Cre/Fbxl5$^{+/+}$ mice injected with poly(I:C) were examined as controls. Deletion of Fbxl5 alleles resulted in increased expression of transferrin receptor 1 (TfR1, also known as CD71) at the cell surface for both hematopoietic progenitors (c-Kit$^+$Sca-1$^+$Lin$^-$, or KSL, cells) and HSCs (CD150$^+$CD48$^-$KSL cells) (Fig. 1a). To examine the intracellular abundance of iron in hematopoietic progenitors, we loaded hematopoietic cells with the iron-sensitive fluorophore calcein-AM, the fluorescence of which is quenched on binding to ferrous iron (Fe$^{2+}$)[20]. The intensity of calcein fluorescence was significantly lower in Fbxl5$^{\Delta/\Delta}$ KSL cells than in control KSL cells (Fig. 1b). Exposure of the cells to the cell-permeable Fe$^{2+}$ chelator 2,2′-bipyridyl abolished the difference in calcein fluorescence intensity between the two genotypes, confirming that the abundance of Fe$^{2+}$ was increased in the Fbxl5$^{\Delta/\Delta}$ KSL cells. These data collectively suggested that FBXL5 governs cellular iron homeostasis in HSCs.

We next examined the role of FBXL5 in the maintenance of HSCs. Flow cytometric analysis revealed a slight but significant reduction in the frequency of KSL cells or HSCs in BM of Fbxl5$^{\Delta/\Delta}$ mice 4 weeks after the last poly(I:C) injection, whereas this reduction was more prominent at 20 weeks (Fig. 1c–e). The frequency of annexin V$^+$ apoptotic cells did not differ between

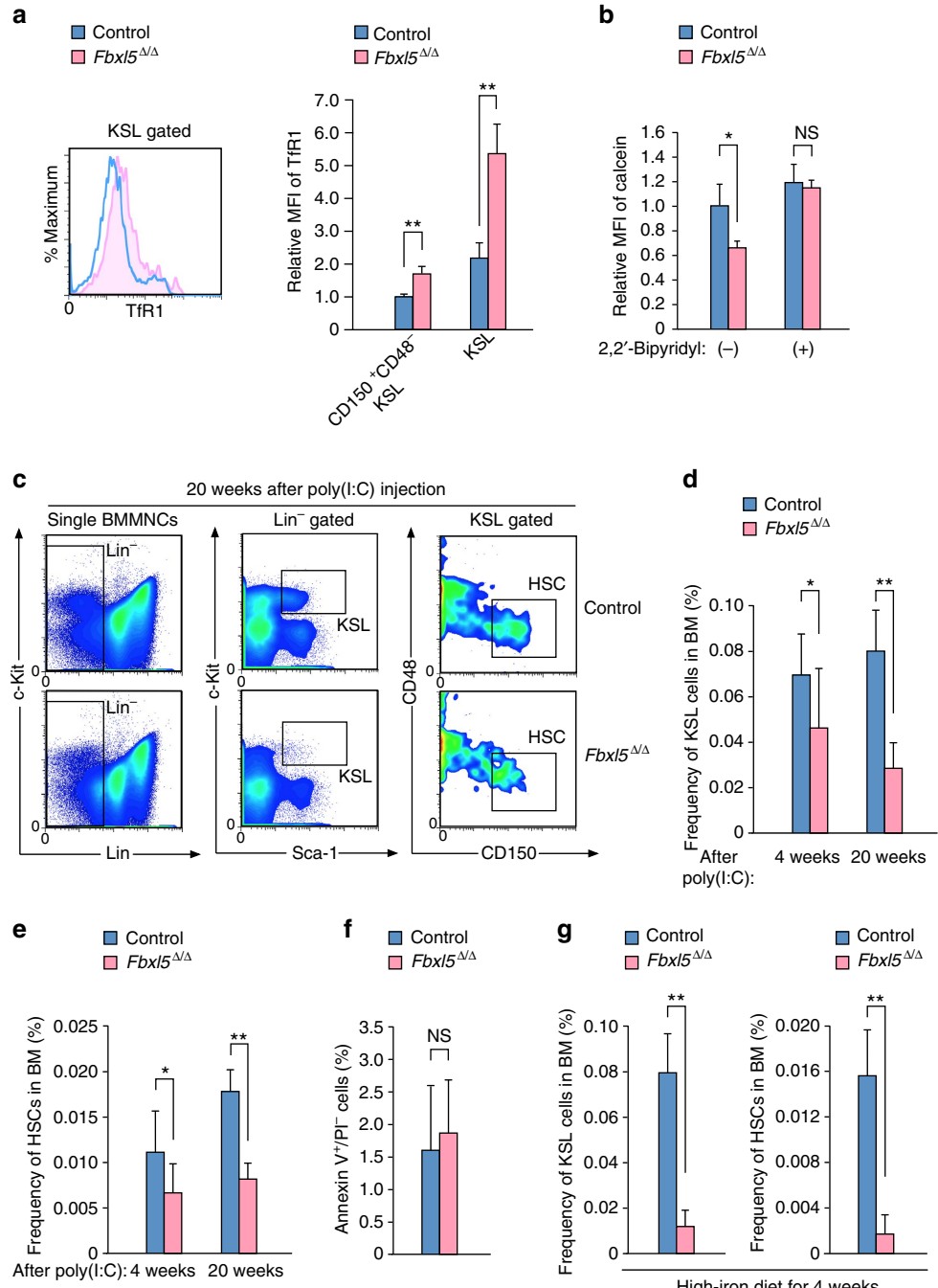

**Figure 1 | Cellular iron homeostasis is essential for the maintenance of HSCs.** (**a**) Representative flow cytometric analysis of TfR1 on the surface of $Fbxl5^{\Delta/\Delta}$ or control KSL cells (left), and relative mean fluorescence intensity (MFI) of TfR1 on the surface of $Fbxl5^{\Delta/\Delta}$ or control KSL cells or HSCs (right, $n=3$). (**b**) Flow cytometric analysis of calcein fluorescence in $Fbxl5^{\Delta/\Delta}$ or control KSL cells exposed or not to 2,2′-bipyridyl ($n=3$). (**c**) Gating strategy for isolation of KSL cells and HSCs from BM mononuclear cells (BMMNCs) of $Fbxl5^{\Delta/\Delta}$ or control mice at 20 weeks after the last poly(I:C) injection. (**d,e**) Frequency of KSL cells (**d**) or HSCs (**e**) among BM cells of $Fbxl5^{\Delta/\Delta}$ or control mice at 4 weeks ($n=9$) or 20 weeks ($n=4$) after the last poly(I:C) injection. (**f**) Frequency of apoptotic [annexin $V^+$/propidium iodide (PI)$^-$] cells among $Fbxl5^{\Delta/\Delta}$ or control HSCs in BM at 4 weeks after the last poly(I:C) injection ($n=3$). (**g**) Frequency of KSL cells or HSCs among BM cells in lethally irradiated mice reconstituted with $Fbxl5^{\Delta/\Delta}$ or control BM cells ($1 \times 10^6$) and fed a high-iron diet for 4 weeks ($n=5$). Data are means + s.d. *$P<0.05$, **$P<0.01$ (Student's $t$-test); NS, not significant.

control and $Fbxl5^{\Delta/\Delta}$ HSCs at 4 weeks (Fig. 1f), indicating that FBXL5 loss impaired HSC maintenance without a significant effect on their survival. These results thus suggested that FBXL5 is required for the maintenance of HSCs.

We further evaluated whether the decrease in the number of $Fbxl5^{\Delta/\Delta}$ hematopoietic progenitors was exacerbated under iron-overload conditions. To avoid a deleterious effect of iron

overload on the liver[4], we transplanted BM cells from either $Mx1$-$Cre$/$Fbxl5^{F/F}$ or $Mx1$-$Cre$/$Fbxl5^{+/+}$ mice not treated with poly(I:C) into lethally irradiated recipients and then injected these animals with poly(I:C). The recipients were fed a high-iron diet for 4 weeks after the last poly(I:C) injection. The frequency of $Fbxl5^{\Delta/\Delta}$ KSL cells or HSCs in BM was found to be markedly reduced compared with that of the corresponding control cells

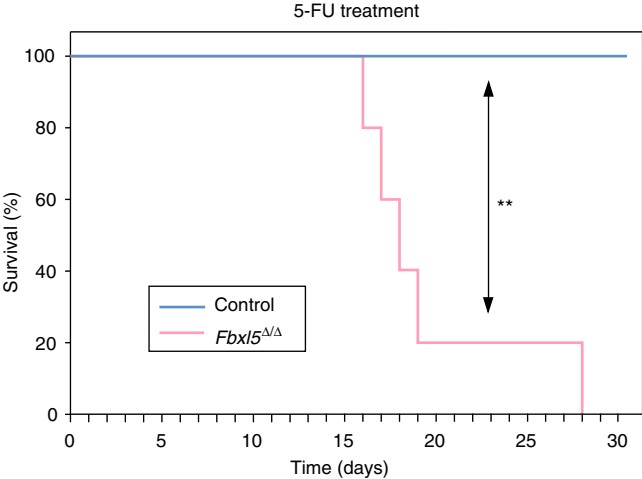

**Figure 2 | Cellular iron overload impairs stress-induced hematopoiesis.** The survival rate of $Fbxl5^{\Delta/\Delta}$ or control mice ($n = 5$) injected with 5-FU (150 mg kg$^{-1}$) at 10-day intervals beginning on day 0 at 20 weeks after the last poly(I:C) injection was determined. **$P < 0.01$ (log-rank test).

after this 4-week period (Fig. 1g), suggesting that FBXL5 deficiency results in exhaustion of HSCs in an iron-dependent manner.

To examine whether the reduced frequency of $Fbxl5^{\Delta/\Delta}$ HSCs in poly(I:C)-treated $Mx1$-$Cre/Fbxl5^{F/F}$ mice compromised hematopoietic capacity *in vivo,* we evaluated the survival rate of the mice after repeated injection with 5-fluorouracil (5-FU) at 10-day intervals to eliminate differentiated hematopoietic cells and induce activation and proliferation of HSCs. The survival rate of the $Fbxl5^{\Delta/\Delta}$ mice began to decline rapidly at $\sim 15$ days after the first 5-FU injection, whereas no control mice died for at least 30 days after the first treatment (Fig. 2), suggesting that stress-induced hematopoiesis is defective in the FBXL5-deficient animals.

We also tested the *in vitro* colony formation capacity of KSL cells in a serial replating assay. Whereas FBXL5 ablation did not affect such capacity at the first plating, the number of clonogenic progenitor cells was markedly reduced for $Fbxl5^{\Delta/\Delta}$ KSL cells at the second and subsequent platings (Fig. 3a), suggesting that this impairment is induced in a cell-autonomous manner. Even at the first plating, iron overload induced by the presence of ferric ammonium citrate (FAC) reduced the colony formation capacity of $Fbxl5^{\Delta/\Delta}$ KSL cells compared with control cells (Fig. 3b). In contrast, $Fbxl5^{\Delta/\Delta}$ and control KSL cells showed a similar colony formation capacity at the second plating in the presence of the ferric iron ($Fe^{3+}$) chelator deferoxamine (DFO). These observations suggested that FBXL5 deficiency results in a deterioration in HSC function due to cellular iron overload.

**Effect of cellular iron overload on differentiated cells.** We next checked whether ablation of FBXL5 in the hematopoietic system affects systemic iron homeostasis and the maintenance of differentiated hematopoietic cells. Measurement of serum iron parameters revealed that the serum iron concentration and transferrin saturation were not altered in $Fbxl5^{\Delta/\Delta}$ mice (Fig. 4a), suggesting that the loss of FBXL5 in the hematopoietic system had no substantial effect on systemic iron homeostasis. Haematologic parameters of $Fbxl5^{\Delta/\Delta}$ mice at 4 weeks after poly(I:C) injection were also similar to those of control mice, with the exception of a slight decrease in the number of platelets (Fig. 4b). We further examined the effect of FBXL5 ablation on the differentiation of hematopoietic cells. The frequency of

differentiated cells (Gr1$^+$ Mac1$^+$ myeloid cells, B220$^+$ B cells or CD3$^+$ T cells) in peripheral blood (PB), BM or the spleen of $Fbxl5^{\Delta/\Delta}$ mice was similar to that in control animals, with the exception of a small increase in the frequency of myeloid cells and decrease in the frequency of B cells in BM (Fig. 4c–e). In addition, erythropoietic parameters of $Fbxl5^{\Delta/\Delta}$ mice did not differ from those of control mice at 20 weeks after poly(I:C) injection (Fig. 4f). The frequency of Ter119$^+$ (erythroid) cells in BM was also not affected by FBXL5 ablation (Fig. 4g). Together, these observations suggested that cellular iron overload induced by FBXL5 ablation had only a small effect on the maintenance of differentiated cells.

**Cellular iron overload impairs HSC self-renewal capacity.** Given that FBXL5 deficiency results in a decline in the number of HSCs, we postulated that FBXL5 is essential for the self-renewal capacity of these cells. To examine this possibility, we first assessed the repopulation capacity of $Fbxl5^{\Delta/\Delta}$ BM cells in a noncompetitive setting. Most lethally irradiated mice transplanted with BM cells ($1 \times 10^6$) from $Fbxl5^{\Delta/\Delta}$ mice died by $\sim 20$ days after BM transfer, whereas those transplanted with control BM cells survived (Fig. 5a), suggesting that the repopulation capacity of $Fbxl5^{\Delta/\Delta}$ BM cells was significantly impaired. To examine the long-term repopulation capacity of $Fbxl5^{\Delta/\Delta}$ HSCs, we performed a competitive reconstitution assay in which $Fbxl5^{\Delta/\Delta}$ or control BM cells were transplanted into lethally irradiated C57BL/6 congenic recipient mice together with competitor cells. Flow cytometric analysis of the resulting chimerism in PB of the recipients until 16 weeks after the BM transfer revealed that the long-term repopulation capacity of $Fbxl5^{\Delta/\Delta}$ HSCs was indeed markedly compromised (Fig. 5b). We further confirmed that $Fbxl5^{\Delta/\Delta}$ hematopoietic cells manifested almost no reconstitution capacity after a second BM transfer (Fig. 5c). The number of KSL cells derived from $Fbxl5^{\Delta/\Delta}$ donor cells was greatly reduced in BM of the initial recipient mice at 16 weeks after BM transfer (Fig. 5d). To characterize the repopulation defect in $Fbxl5^{\Delta/\Delta}$ HSCs, we examined the homing capacity of $Fbxl5^{\Delta/\Delta}$ hematopoietic progenitor cells after transplantation. $Fbxl5^{\Delta/\Delta}$ or control KSL cells were sorted, labelled with carboxyfluorescein succinimidyl ester (CFSE) and transplanted into lethally irradiated recipients, and the recipient BM was analysed 16 h after transplantation. The homing capacity of the transplanted CFSE$^+$ cells for BM was similar for the two genotypes (Fig. 5e), excluding the possibility that a homing defect is responsible for the repopulation defect of $Fbxl5^{\Delta/\Delta}$ HSCs. To evaluate the stem cell capacity of $Fbxl5^{\Delta/\Delta}$ HSCs excluding homing and engraftment, we transplanted BM cells from either $Mx1$-$Cre/Fbxl5^{F/F}$ or $Mx1$-$Cre/Fbxl5^{+/+}$ mice not treated with poly(I:C) into lethally irradiated recipients together with competitor cells. Four weeks after BM transfer, we confirmed that donor cells were reconstituted in the recipient BM and then injected the recipient mice with poly(I:C). $Fbxl5^{\Delta/\Delta}$ HSCs gradually lost long-term repopulation capacity (Fig. 5f), showing that such capacity after homing and engraftment was impaired in $Fbxl5^{\Delta/\Delta}$ HSCs. The differentiation of $Fbxl5^{\Delta/\Delta}$ HSCs appeared normal in this setting of competitive repopulation (Fig. 5g). These results collectively indicated that cellular iron homeostasis is essential for the self-renewal capacity of HSCs.

**Role of IRP2 in the effect of FBXL5 on HSC stemness.** Given that impaired degradation of IRP2 is primarily responsible for the embryonic mortality of $Fbxl5^{-/-}$ mice[4], we hypothesized that the defective stem cell capacity of $Fbxl5^{\Delta/\Delta}$ HSCs might also be due to IRP2 accumulation. We therefore evaluated whether suppression of IRP2 is required for the repopulation capacity of HSCs with the

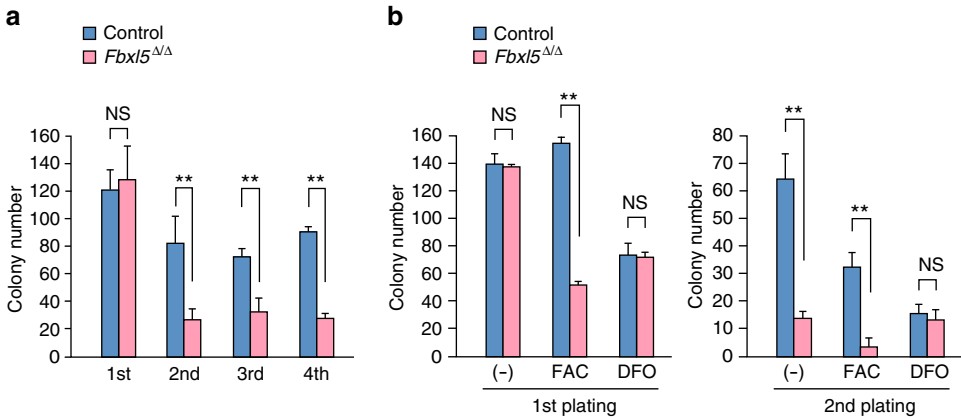

**Figure 3 | Cellular iron overload impairs colony formation capacity of hematopoietic progenitors *in vitro*.** (**a**) Serial replating assay of colony formation by *Fbxl5*$^{\Delta/\Delta}$ or control KSL cells ($n = 6$). (**b**) Serial replating assay of colony formation by *Fbxl5*$^{\Delta/\Delta}$ or control KSL cells in the absence or presence of FAC (100 µg ml$^{-1}$) or 10 µM DFO ($n = 3$). Data are means + s.d. **$P < 0.01$ (Student's $t$-test).

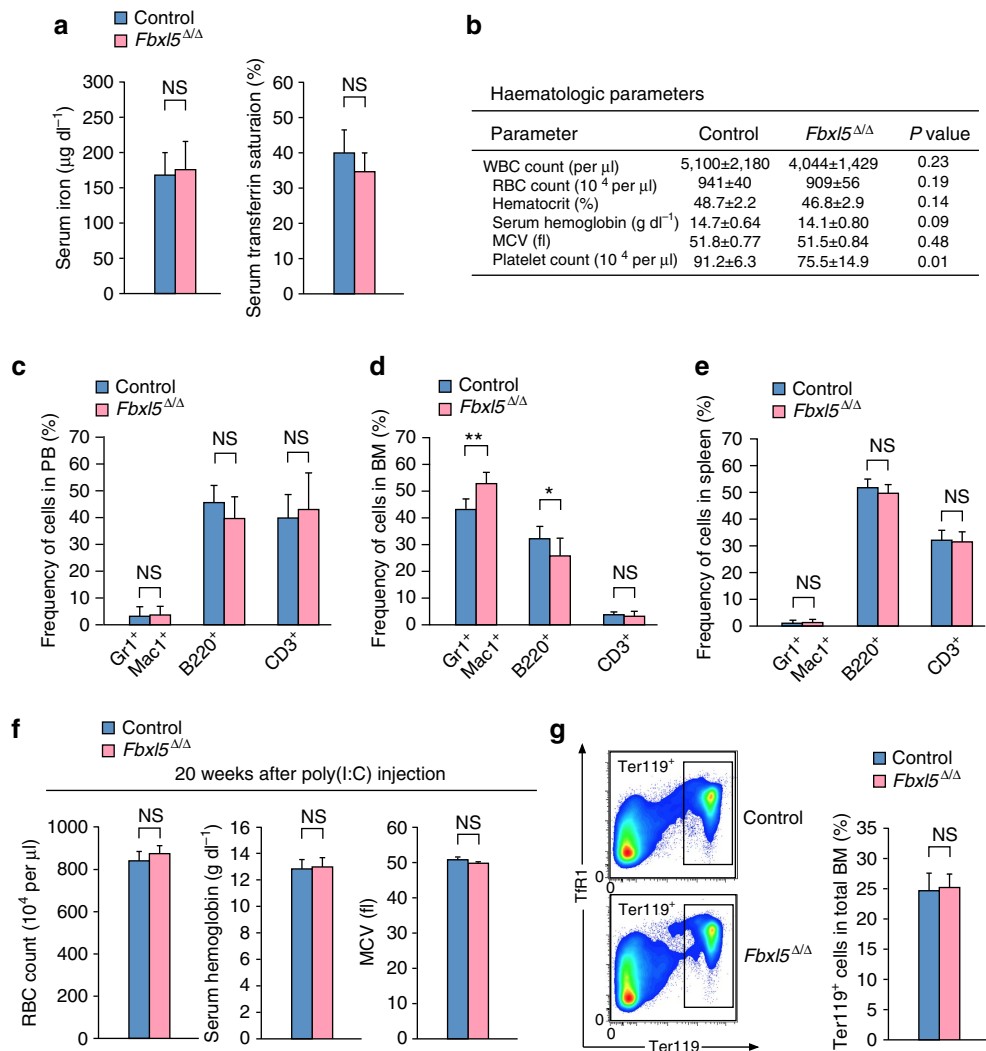

| Parameter | Control | *Fbxl5*$^{\Delta/\Delta}$ | $P$ value |
|---|---|---|---|
| WBC count (per µl) | 5,100±2,180 | 4,044±1,429 | 0.23 |
| RBC count ($10^4$ per µl) | 941±40 | 909±56 | 0.19 |
| Hematocrit (%) | 48.7±2.2 | 46.8±2.9 | 0.14 |
| Serum hemoglobin (g dl$^{-1}$) | 14.7±0.64 | 14.1±0.80 | 0.09 |
| MCV (fl) | 51.8±0.77 | 51.5±0.84 | 0.48 |
| Platelet count ($10^4$ per µl) | 91.2±6.3 | 75.5±14.9 | 0.01 |

**Figure 4 | Cellular iron overload has only a small effect on differentiated cells.** (**a**) Serum iron concentration (left) and transferrin saturation (right) in *Fbxl5*$^{\Delta/\Delta}$ or control mice ($n = 9$). (**b**) Haematologic parameters in *Fbxl5*$^{\Delta/\Delta}$ or control mice ($n = 9$). WBC, white blood cell; RBC, red blood cell; MCV, mean corpuscular volume. (**c–e**) Frequency of differentiated cells (Gr1$^+$Mac1$^+$ myeloid cells, B220$^+$ B cells and CD3$^+$ T cells) in PB ($n = 9$) (**c**), BM ($n = 9$) (**d**) and the spleen ($n = 6$) (**e**) of *Fbxl5*$^{\Delta/\Delta}$ or control mice. (**f**) Haematologic parameters related to erythropoiesis in *Fbxl5*$^{\Delta/\Delta}$ or control mice at 20 weeks after the last poly(I:C) injection ($n = 4$). (**g**) Representative flow cytometric analysis of Ter119 on the surface of BM cells as well as the frequency of Ter119$^+$ erythroid cells in BM for *Fbxl5*$^{\Delta/\Delta}$ or control mice ($n = 3$) at 4 weeks after poly(I:C) injection. Data are means + s.d. *$P < 0.05$, **$P < 0.01$ (Student's $t$-test); NS, not significant.

use of a competitive reconstitution assay. Both the repopulation and differentiation capacities of Irp2[−/−] HSCs were similar to those of control (Irp2[+/+]) HSCs (Fig. 6a–c). We next prepared Mx1-Cre/Fbxl5[F/F]/Irp2[−/−] and Mx1-Cre/Fbxl5[+/+]/Irp2[+/+] mice to be able to analyse Fbxl5[Δ/Δ]/Irp2[−/−] HSCs after poly(I:C)

injection. The competitive reconstitution assay revealed that the long-term repopulation capacity of Fbxl5[Δ/Δ]/Irp2[−/−] HSCs did not differ significantly from that of control HSCs after the first or second BM transfer (Fig. 6d–f), suggesting that aberrant IRP2 activity is responsible for the deleterious effect of FBXL5

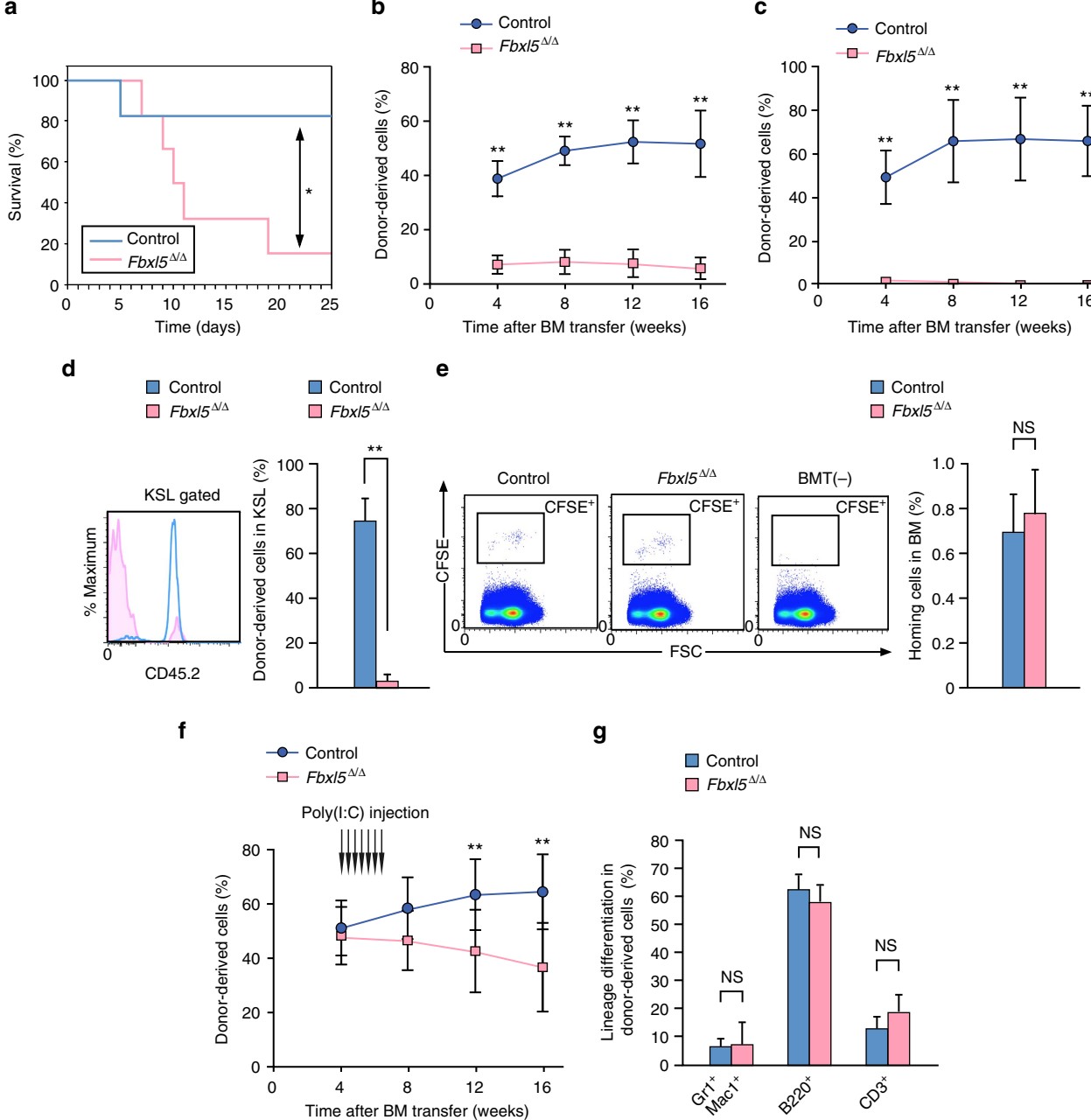

**Figure 5 | Cellular iron overload impairs the repopulation capacity of HSCs.** (**a**) Survival of lethally irradiated mice transplanted with BM cells ($1 \times 10^6$) from Fbxl5[Δ/Δ] or control mice ($n = 6$). (**b**) Hematopoietic repopulation capacity of Fbxl5[Δ/Δ] or control BM cells ($4 \times 10^5$) transplanted together with an equal number of competitor cells. The percentage of donor-derived cells in PB of recipient mice was determined at the indicated times after BM transfer from Fbxl5[Δ/Δ] ($n = 7$) or control ($n = 8$) mice. (**c**) BM cells ($1 \times 10^6$) from the recipient mice in **b** at 16 weeks after BM transfer were serially transplanted into additional recipient mice (Fbxl5[Δ/Δ], $n = 6$; control, $n = 5$), and the percentage of donor-derived cells in PB of the new recipients was determined at the indicated times thereafter. (**d**) The percentage of donor-derived cells among KSL cells in BM of the recipients in **b** was determined at 16 weeks after BM transfer. (**e**) Homing capacity of CFSE-labelled Fbxl5[Δ/Δ] or control KSL cells ($2 \times 10^4$) for BM at 16 h after BM transfer (BMT) ($n = 3$). FSC, forward scatter. (**f**) Irradiated recipient mice were transplanted with donor BM cells ($4 \times 10^5$) from Mx1-Cre/Fbxl5[+/+] (control) or Mx1-Cre/Fbxl5[F/F] mice [not injected with poly(I:C)] together with an equal number of competitor BM cells, and they were injected with poly(I:C) beginning 4 weeks after BM transfer. The percentage of donor-derived cells in PB at the indicated times after BM transfer was determined (Fbxl5[Δ/Δ], $n = 9$; control, $n = 10$). (**g**) Frequency of differentiated cells (Gr1[+] Mac1[+] myeloid cells, B220[+] B cells, and CD3[+] T cells) among donor-derived blood cells for the recipients in **f** at 16 weeks after BM transfer. Data are means ± s.d. *$P < 0.05$, **$P < 0.01$ versus the corresponding value for Fbxl5[Δ/Δ] or as indicated [log-rank test (**a**) or Student's $t$-test (other panels)].

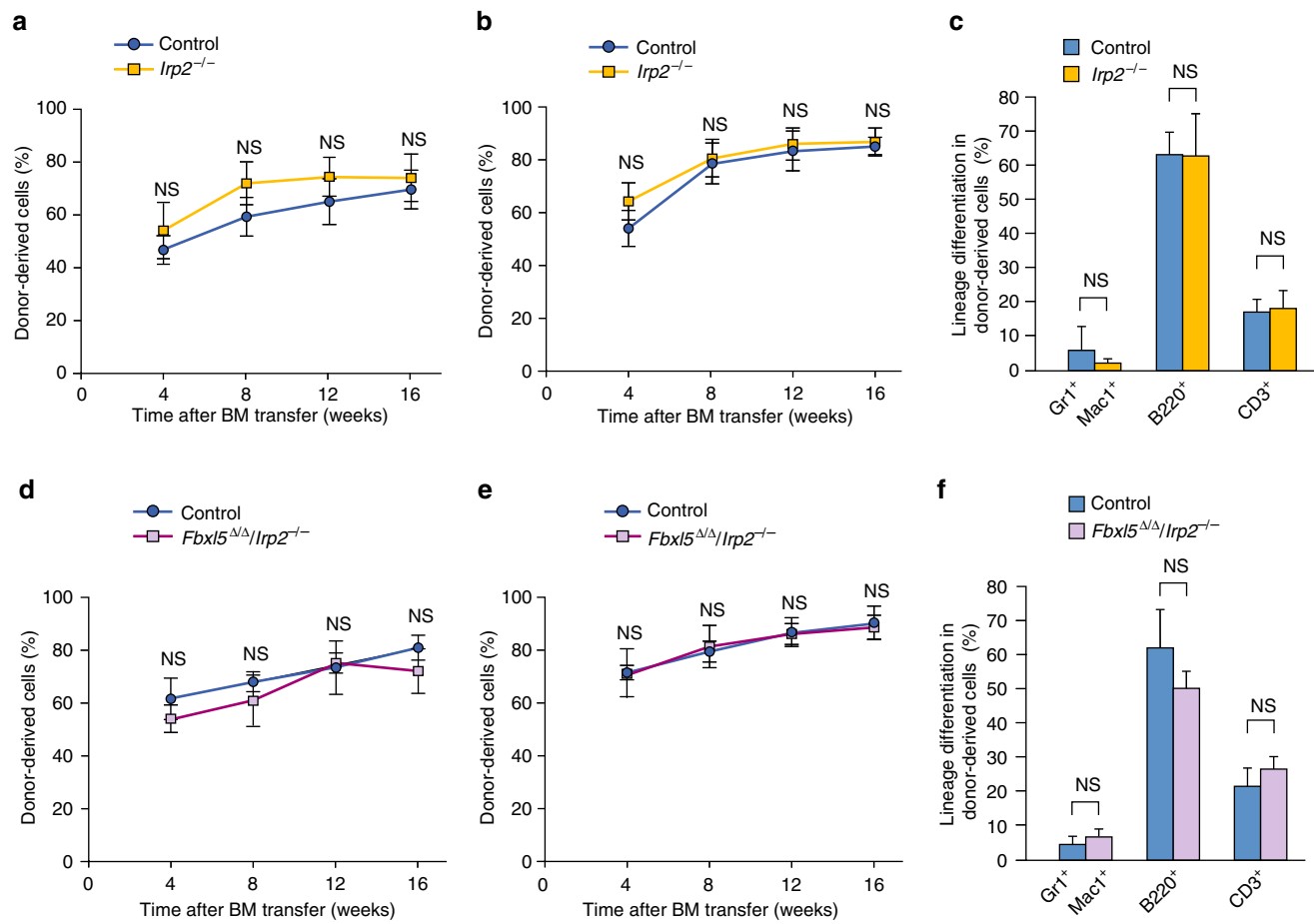

**Figure 6 | Suppression of IRP2 mediates the effect of FBXL5 on the repopulation capacity of HSCs.** (**a**) Hematopoietic repopulation capacity of $Irp2^{-/-}$ or control ($Irp2^{+/+}$) BM cells ($4 \times 10^5$) transplanted together with an equal number of competitor cells. The percentage of donor-derived cells in PB of recipient mice was determined at the indicated times after BM transfer ($Irp2^{-/-}$, $n = 5$; control, $n = 4$). (**b**) BM cells ($1 \times 10^6$) from the recipient mice in **a** at 16 weeks after BM transfer were serially transplanted into additional recipient mice ($n = 5$), and the percentage of donor-derived cells in PB of the new recipients was determined at the indicated times thereafter. (**c**) Frequency of differentiated cells (Gr1$^+$Mac1$^+$ myeloid cells, B220$^+$ B cells and CD3$^+$ T cells) in donor-derived blood cells of the recipients in **a** at 16 weeks after BM transfer. (**d**) Hematopoietic repopulation capacity of $Fbxl5^{\Delta/\Delta}/Irp2^{-/-}$ or control ($Fbxl5^{+/+}/Irp2^{+/+}$) BM cells ($4 \times 10^5$) transplanted together with an equal number of competitor cells. The percentage of donor-derived cells in PB of recipient mice was determined at the indicated times after BM transfer ($n = 4$). (**e**) BM cells ($1 \times 10^6$) from the recipient mice in **d** at 16 weeks after BM transfer were serially transplanted into additional recipient mice ($n = 6$), and the percentage of donor-derived cells in PB of the new recipients was determined at the indicated times thereafter. (**f**) Frequency of differentiated cells (Gr1$^+$Mac1$^+$ myeloid cells, B220$^+$ B cells and CD3$^+$ T cells) in donor-derived blood cells of the recipients in **d** at 16 weeks after BM transfer. Data are means ± s.d. NS for comparison of corresponding values for the pairs of genotypes (Student's $t$-test).

ablation on the repopulation capacity of HSCs. FBXL5 thus protects HSC stemness through suppression of IRP2.

**Cellular iron overload disrupts redox regulation in HSCs.** To investigate further the mechanism by which cellular iron overload impairs HSC function, we profiled gene expression in $Fbxl5^{\Delta/\Delta}$ HSCs. Microarray analysis identified 1,128 differentially expressed genes (686 downregulated and 442 upregulated; fold change of $> 1.5$ or $< -1.5$ and $P$ value of $< 0.05$) in $Fbxl5^{\Delta/\Delta}$ HSCs compared with control cells (Supplementary Data 1 and 2). A complete list of these genes has been deposited in the Gene Expression Omnibus (GEO) database under the accession number GSE93649. As expected, expression of genes related to cellular iron metabolism such as *Hbb*, *Slc48a1*, *Ftl1*, *Lcn2* and *Abcb6* was upregulated. The upregulated genes also included many genes important for redox regulation, such as *Mt1*, *Mt2*, *Hmox1*, *Gstm2*, *Slc7a11*, *Gclm*, *Gsta4*, *Cat*, *Txn1*, *Nqo1* and *Sod1*.

Ingenuity pathway analysis (IPA) revealed that the differentially expressed genes were most highly associated with the NRF2-mediated oxidative stress response (Fig. 7a). Gene set enrichment analysis (GSEA) also confirmed that the antioxidant defense system is activated in $Fbxl5^{\Delta/\Delta}$ HSCs (Fig. 7b,c). Changes in the expression of several genes related to oxidative stress responses were validated by RT and rtPCR analysis (Fig. 7d). On the basis of these results, we concluded that FBXL5 ablation evokes oxidative stress in HSCs. Oxidative stress in HSCs is also known to give rise to phosphorylation (activation) of p38 mitogen-activated protein kinase (MAPK), which results in aberrant cell proliferation and exhaustion[1,2]. Intracellular flow cytometric analysis revealed that the frequency of cells positive for phosphorylated p38 MAPK was greater among $Fbxl5^{\Delta/\Delta}$ HSCs than among control HSCs (Fig. 7e). The mean fluorescence intensity (MFI) of phospho-p38 for $Fbxl5^{\Delta/\Delta}$ HSCs also tended to be greater than that for control HSCs, although this difference did not achieve statistical significance ($P = 0.062$). Given that both extrinsic

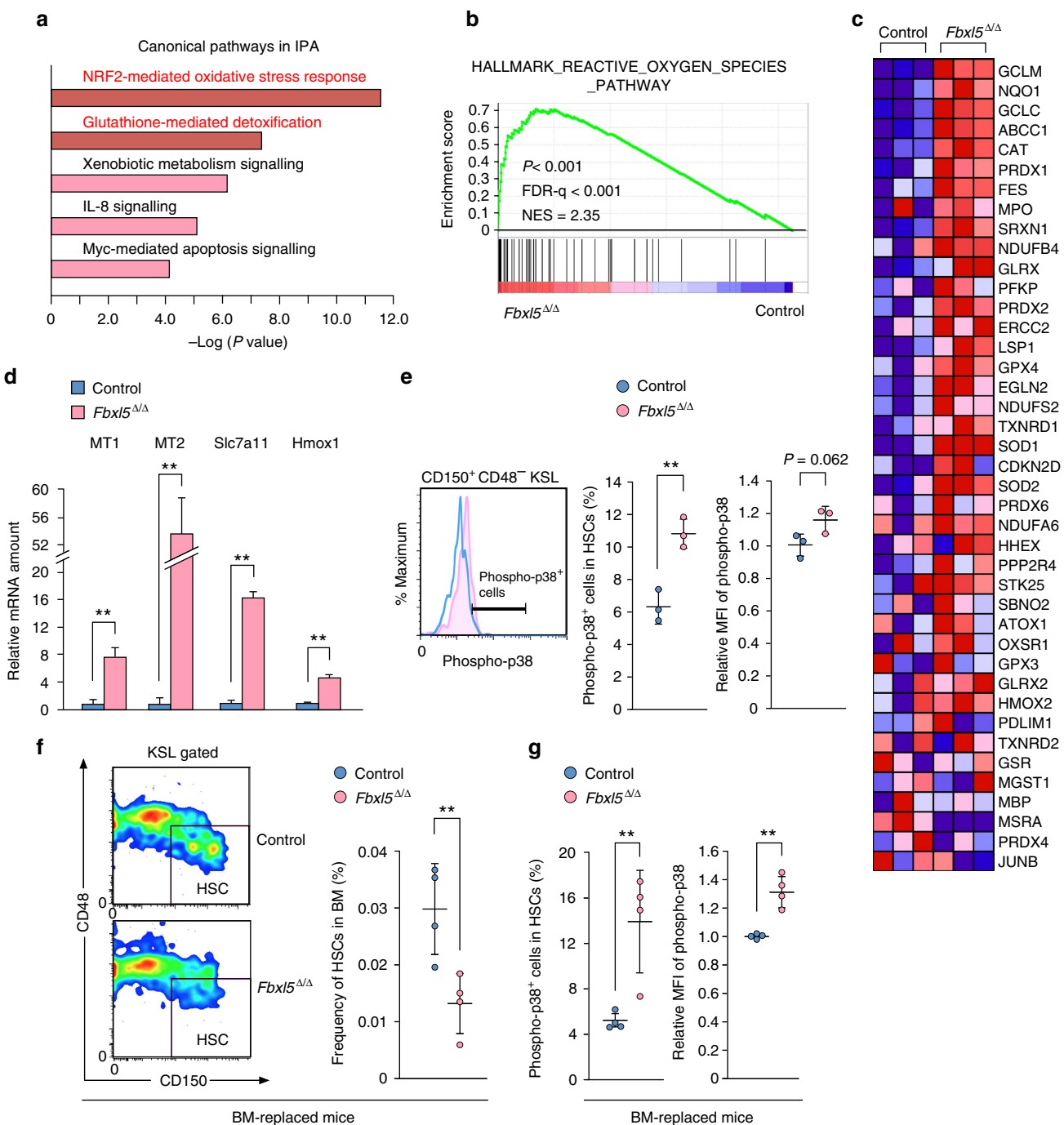

**Figure 7 | Cellular iron overload disrupts redox regulation in HSCs. (a)** Top-ranked canonical pathways for dysregulated genes in *Fbxl5*$^{\Delta/\Delta}$ HSCs as revealed by IPA. Genes with a *P* value of <0.05 and a fold change in expression of >1.3 or <−1.3 were analysed. **(b)** GSEA plot of differentially expressed genes for the list of genes related to the ROS pathway (*n* = 3). FDR-*q*, false discovery rate *q* value; NES, normalized enrichment score. **(c)** Heat map for genes in the ROS pathway in **b**. **(d)** RT and real-time PCR analysis of mRNAs for ROS-related genes in *Fbxl5*$^{\Delta/\Delta}$ or control HSCs (*n* = 3). **(e)** Flow cytometric analysis of the frequency of cells positive for phosphorylated p38 MAPK (*n* = 3) and relative MFI of phospho-p38 (*n* = 3) among *Fbxl5*$^{\Delta/\Delta}$ or control HSCs. **(f,g)** Frequency of HSCs among BM cells **(f)** as well as frequency of cells positive for phosphorylated p38 MAPK and relative MFI of phospho-p38 among HSCs in BM **(g)** of lethally irradiated mice at 4 weeks after reconstitution with *Mx1-Cre/Fbxl5*$^{F/F}$ or *Mx1-Cre/Fbxl5*$^{+/+}$ (control) BM cells and injected with poly(I:C) (*n* = 4). Data are means ± s.d. **P < 0.01 (Student's *t*-test).

factors including various cytokines as well as intrinsic factors such as oxidative stress influence p38 MAPK phosphorylation status[21], we sought to examine the influence of only intrinsic factors on p38 phosphorylation in HSCs. To this end, we transplanted BM cells from *Mx1-Cre/Fbxl5*$^{F/F}$ or *Mx1-Cre/Fbxl5*$^{+/+}$ mice into lethally irradiated recipients and

then injected these animals with poly(I:C). The frequency of *Fbxl5*$^{\Delta/\Delta}$ HSCs in BM of the recipients was significantly reduced compared with that of control HSCs (Fig. 7f), as was the case for *Mx1-Cre/Fbxl5*$^{F/F}$ mice treated with poly(I:C) (Fig. 1e). The frequency of cells positive for phosphorylated p38 MAPK as well as the MFI for phospho-p38 were also significantly

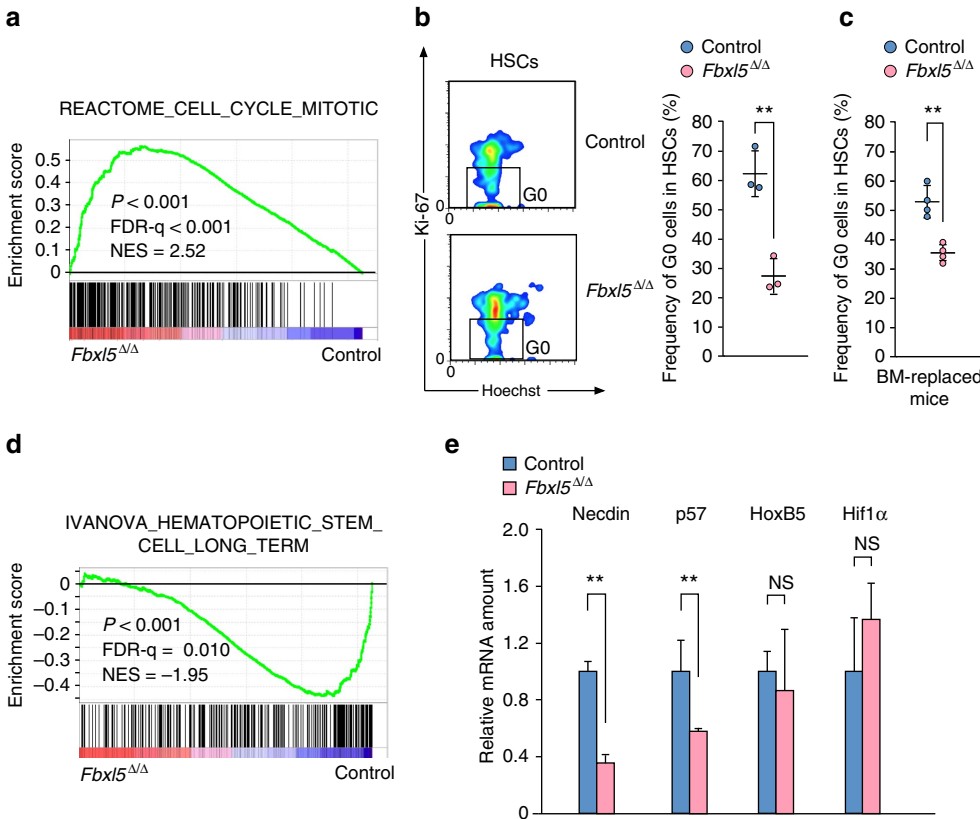

**Figure 8 | Cellular iron homeostasis is essential for dormancy and a stem cell signature in HSCs.** (**a**) GSEA plot of differentially expressed genes in $Fbxl5^{\Delta/\Delta}$ or control HSCs for the list of genes related to proliferation. (**b**) Flow cytometric analysis of cell cycle status by staining with antibodies to Ki-67 and Hoechst 33342 for determination of the percentage of cells in $G_0$ phase among $Fbxl5^{\Delta/\Delta}$ or control HSCs ($n = 3$). (**c**) Flow cytometric analysis of cell cycle status for HSCs in BM of lethally irradiated mice reconstituted with $Fbxl5^{\Delta/\Delta}$ or control BM cells ($n = 4$). (**d**) GSEA plot of differentially expressed genes in $Fbxl5^{\Delta/\Delta}$ or control HSCs for the list of HSC-specific genes. (**e**) RT and real-time PCR analysis of mRNAs for HSC-specific genes in $Fbxl5^{\Delta/\Delta}$ or control HSCs ($n = 3$). Data are means ± s.d. $**P < 0.01$ (Student's $t$-test).

increased in $Fbxl5^{\Delta/\Delta}$ HSCs compared with control HSCs in the recipient mice (Fig. 7g). These results thus also confirmed that $Fbxl5^{\Delta/\Delta}$ HSCs are exposed to intense oxidative stress.

**Cellular iron homeostasis is essential for HSC dormancy.** GSEA also showed that the gene expression profile of $Fbxl5^{\Delta/\Delta}$ HSCs was shifted toward proliferation compared with that of control HSCs (Fig. 8a). Given that the loss of dormancy in HSCs leads to their exhaustion, we hypothesized that FBXL5 ablation might promote exit from the dormant state in HSCs. We therefore evaluated the cell cycle kinetics of $Fbxl5^{\Delta/\Delta}$ HSCs by intracellular staining of the proliferation marker Ki-67 and analysis of DNA ploidy by staining with Hoechst 33342. The frequency of cells in the dormant state (Ki-67$^-$ fraction) was reduced for $Fbxl5^{\Delta/\Delta}$ HSCs compared with control cells (Fig. 8b). A similar difference in the frequency of dormant HSCs was also observed in poly(I:C)-treated recipients of transplanted BM cells from $Mx1$-$Cre/Fbxl5^{F/F}$ or $Mx1$-$Cre/Fbxl5^{+/+}$ mice (Fig. 8c). GSEA also revealed loss of an HSC-specific gene signature in $Fbxl5^{\Delta/\Delta}$ HSCs (Fig. 8d). The top 30 downregulated genes in our microarray analysis (Supplementary Data 2) include the HSC-specific gene $Necdin$[22], whose downregulation in $Fbxl5^{\Delta/\Delta}$ HSCs was confirmed by RT and rtPCR analysis (Fig. 8e). We also confirmed the downregulation of the HSC-specific gene $p57$ (ref. 23), whereas the expression of other such genes including $Hoxb5$ (ref. 24) and $Hif1a$ (ref. 25) was not affected (Fig. 8e). These results suggested that cellular iron overload induces exit of HSCs from the dormant state and loss of an HSC-specific gene expression signature.

**Disrupted iron homeostasis in HSCs is associated with MDS.** The results of our mouse experiments together indicated that cellular iron homeostasis governed by FBXL5 plays an essential role in the maintenance and function of HSCs. We finally examined whether FBXL5 deficiency might be associated with human hematopoietic diseases. Impaired HSC function can result in the development of MDS, a clonal HSC disorder characterized by ineffective hematopoiesis[9–11]. Expression of $FBXL5$ was shown to be differentially downregulated in CD34$^+$CD38$^-$CD90$^+$ HSCs from MDS patients with deletion of chromosome 5q relative to those from healthy control subjects[26]. Analysis of a published set of microarray data revealed that $FBXL5$ expression was also significantly downregulated in Lin$^-$CD34$^+$CD38$^-$CD90$^+$CD45RA$^-$ HSCs from eight MDS patients without deletion of chromosome 5q compared with 11 age-matched healthy control samples[27] (Fig. 9a). It is of note that the expression of both $TFR1$ and $DMT1$, which is upregulated by IRP2 and therefore represents an index of IRP2 activity, was also increased in the MDS patients (Fig. 9b,c). To determine whether $FBXL5$ expression is also downregulated in more differentiated hematopoietic progenitor cells in MDS patients, we evaluated a published set of microarray data for CD34$^+$ hematopoietic progenitor cells from 183 MDS patients with various cytogenetic abnormalities and 17 healthy control subjects[28]. Expression of $FBXL5$ was significantly downregulated in the CD34$^+$ cells of patients with refractory anaemia with ringed sideroblasts (RARS), a subgroup of MDS characterized by iron deposition and apoptosis in hematopoietic progenitor cells (Fig. 9d). Consistent

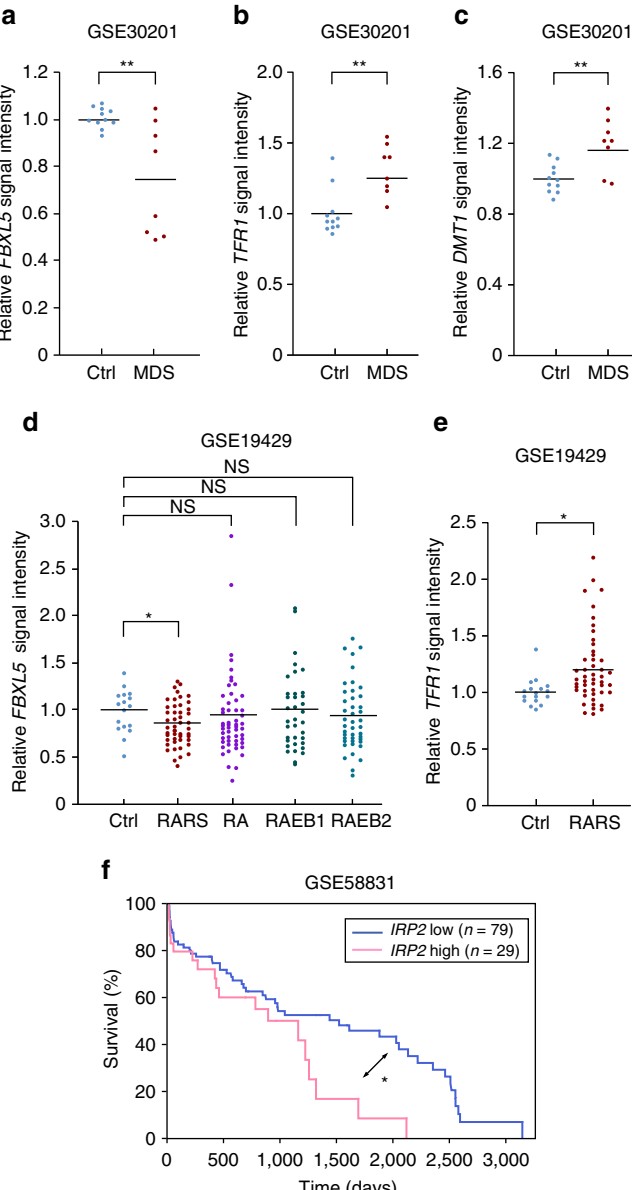

**Figure 9 | Downregulation of *FBXL5* expression is associated with human hematopoietic failure.** (a–c) Microarray analysis of *FBXL5* (a), *TFR1* (b) and *DMT1* (c) expression in Lin⁻CD34⁺CD38⁻CD90⁺CD45RA⁻ cells from healthy control (Ctrl) subjects ($n = 11$) or from MDS patients without deletion of chromosome 5q ($n = 8$). (d,e) Microarray analysis of *FBXL5* (d) and *TFR1* (e) expression in CD34⁺ BM mononuclear cells from healthy control subjects ($n = 17$) or from patients with refractory anaemia with ringed sideroblasts (RARS, $n = 48$), refractory anaemia (RA, $n = 55$), refractory anaemia with excess blasts 1 (RAEB1, $n = 37$), or refractory anaemia with excess blasts 2 (RAEB2, $n = 43$). Each point represents an individual donor, and horizontal lines indicate the mean. (f) Survival of MDS patients without deletion of chromosome 5q and with a high ($n = 29$) or low ($n = 79$) level of *IRP2* expression in CD34⁺ hematopoietic progenitor cells. *$P < 0.05$, **$P < 0.01$ (Student's *t*-test (a–e) or log-rank test (f)).

with this finding, *TFR1* expression was also significantly upregulated in the CD34⁺ cells from the RARS patients (Fig. 9e). These findings implicate cellular iron overload due to FBXL5 downregulation in the pathogenesis of human hematopoietic failure. Given that suppression of aberrant IRP2 activity cancelled the deleterious effect of FBXL5 ablation on

the repopulation capacity of HSCs (Fig. 6), IRP2 is a potential therapeutic target for cellular iron overload in HSCs due to FBXL5 downregulation, including that in patients with MDS. Consistent with this notion, a published microarray data set revealed that an increased IRP2 mRNA abundance in CD34⁺ hematopoietic progenitor cells was related to reduced survival in MDS patients without deletion of chromosome 5q (ref. 29) (Fig. 9f).

## Discussion

We have here discovered a previously unrecognized role for cellular iron homeostasis in the maintenance of HSCs with the use of mouse models of conditional *Fbxl5* deletion. Mechanistically, cellular iron homeostasis in HSCs regulates oxidative stress, quiescence and self-renewal capacity. Analysis of public data sets revealed that downregulation of *FBXL5* expression was associated with MDS, a disease characterized by BM failure. Suppression of IRP2 activity in FBXL5-deficient HSCs restored stem cell function, implicating IRP2 as a potential therapeutic target for cellular iron overload in HSCs with FBXL5 deficiency (Fig. 10).

FBXL5 is a master regulator of cellular iron metabolism by virtue of its role as the substrate recognition component of the SCF^FBXL5 E3 ubiquitin ligase for IRP2 degradation[5,6]. Other proteins targeted by FBXL5 for proteasomal degradation include p150^Glued, cortactin and single-stranded DNA binding protein 1 (SSB1)[30–32]. FBXL5 has also been shown to interact with Snail1 (refs 33,34) and CBP/p300-interacting transactivator 2 (CITED2)[35], leading to their degradation. CITED2 was shown to control the proliferation of mouse embryonic fibroblasts by promoting expression of the Polycomb group genes *Bmi1* and *Mel18* (ref. 36) as well as to selectively maintain adult HSC function at least in part through regulation of p16 and p53 (ref. 37). If CITED2 accumulates in FBXL5-deficient HSCs, it might also promote their proliferation and exhaustion. However, FBXL5-deficient mice die during embryogenesis and their mortality is prevented by additional ablation of IRP2, suggesting that impaired IRP2 degradation is primarily responsible for the embryonic death[4]. Our data now provide evidence that IRP2 is also the major target of SCF^FBXL5 in HSCs, given that the defect in repopulation capacity of FBXL5-deficient HSCs was rescued by additional ablation of IRP2. These lines of evidence also suggest that the contribution of CITED2 to the phenotype of FBXL5-deficient HSCs is limited. We therefore conclude that FBXL5 plays an essential role in the maintenance of HSCs through suppression of IRP2 activity.

Our present study shows that disruption of cellular iron homeostasis by FBXL5 ablation in HSCs resulted in cellular iron overload, oxidative stress responses, exit from dormancy and eventual exhaustion in these cells. Increased ROS levels promote the proliferation and differentiation of HSCs, primarily via modulation of p38 MAPK and the forkhead box O (FOXO) family of transcription factors[1]. Many mutations that result in aberrantly high ROS levels in HSCs also lead to impairment of quiescence and self-renewal potential as a result of enhanced differentiation[38]. We now show that FBXL5 is also an essential ROS regulator in HSCs and plays a key role in the maintenance of stemness. The expression of many genes that contribute to oxidative stress responses was found to be upregulated in FBXL5-deficient HSCs: The most upregulated ROS-related genes included those for metallothionein (MT) 1 and MT2, which are small, cysteine-rich, and heavy metal-binding proteins that participate in an array of protective stress responses[39]. These proteins thus protect cells from exposure to oxidants and electrophiles, which react readily with sulfhydryl groups.

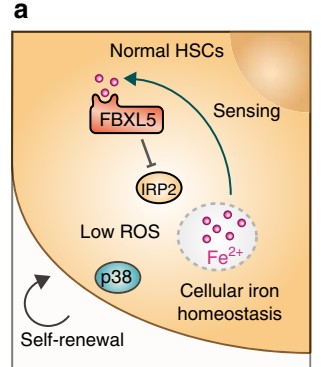
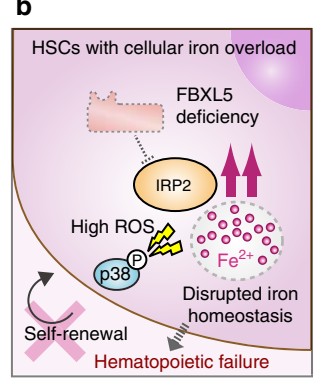
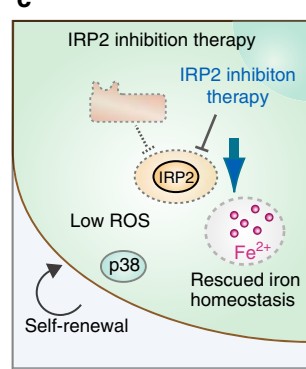

**Figure 10 | Cellular iron homeostasis is essential for the maintenance of HSCs.** (**a**) Limitation of the production of ROS is essential for the maintenance of HSCs. Iron is a major elicitor of oxidative stress, with the result that cellular iron homeostasis is strictly regulated in HSCs. Cellular iron homeostasis is governed by the FBXL5–IRP2 axis in these cells. (**b**) FBXL5 deficiency in HSCs results in cellular iron overload as a result of IRP2 overactivity. Cellular iron overload promotes oxidative stress, which leads to dysfunction and exhaustion of HSCs. Downregulation of *FBXL5* expression in HSCs is apparent in human hematopoietic failure such as that associated with MDS. (**c**) Suppression of aberrant IRP2 activity attenuates the deleterious effect of FBXL5 ablation on the stem cell capacity of HSCs, suggesting that IRP2 is a potential therapeutic target for cellular iron overload in HSCs associated with FBXL5 downregulation.

Moreover, they play a key role in regulation of cellular zinc levels by binding and releasing zinc. The marked upregulation of MT gene expression in FBXL5-deficient HSCs might thus also modify cellular zinc metabolism. Another such upregulated gene was *Slc7a11*, which encodes a component of system Xc⁻. System Xc⁻ contributes to the maintenance of redox homeostasis by importing cystine for synthesis of the major cellular antioxidant glutathione. Inhibition of system Xc⁻ by erastin in cancer cells triggers ferroptosis, a recently recognized form of iron-dependent cell death[40]. Upregulation of *Slc7a11* expression might therefore represent a mechanism to protect HSCs with cellular iron overload from ferroptosis. Indeed, FBXL5 ablation was shown to promote erastin-induced ferroptosis in cultured cells[40]. These changes in gene expression in FBXL5-deficient HSCs are thus indicative of iron-mediated cellular damage and disruption of redox homeostasis.

In contrast to the deleterious effect of iron overload evoked by FBXL5 loss on HSC function, FBXL5 deficiency did not substantially affect differentiated hematopoietic cells. Although cross talk between systemic iron homeostasis and erythropoiesis is well established[41], detailed analysis of erythropoiesis in FBXL5-deficient mice indicated that FBXL5 has a limited role in erythropoiesis, consistent with the finding that the amount of FBXL5 mRNA is smallest in the erythroid lineage among differentiated haematopoietic cells. In general, erythroid cells require large amounts of iron to sustain haemoglobin synthesis[41], suggesting that the importance of FBXL5 as a brake on iron uptake might be rather limited in erythropoiesis.

Systemic iron overload is sometimes a complication of hematopoietic failure such as that associated with MDS as a result of the required frequent blood transfusions and the suppression of hepcidin production due to ineffective erythropoiesis[12]. Systemic iron overload in turn has a suppressive effect on hematopoiesis in patients with hematopoietic failure, with iron-chelation therapy having been found to be beneficial in these patients[13–15]. However, iron overload has little effect on hematopoiesis in patients with hereditary hemochromatosis, which is also a major cause of systemic iron overload due to hepcidin deficiency. These various observations suggest that the hematopoietic system in some patients with BM failure is intrinsically vulnerable to iron. By analysing public data sets, we found that *FBXL5* expression was significantly downregulated in HSCs in a subset of patients with MDS. Reduced expression of *FBXL5* might promote loss of

quiescence and exhaustion in HSCs of such patients. Similar to the *Fbxl5*^Δ/Δ HSCs examined in the present study, HSCs of MDS patients with low *FBXL5* expression might also be vulnerable to systemic iron overload. The expression of *FBXL5* was also found to be significantly downregulated in CD34⁺ progenitor cells of patients with RARS, a subtype of MDS characterized by iron deposition in hematopoietic progenitor cells. Downregulation of *FBXL5* expression in RARS is of interest given that RARS progenitor cells are loaded with excess iron in mitochondria and are vulnerable to ROS-induced apoptosis[9]. FBXL5 deficiency might thus exacerbate the cellular and mitochondrial iron overload in RARS progenitor cells, contributing to disease pathogenesis. Our findings raise the possibility that FBXL5 plays a key role in the pathogenesis of BM failure syndromes including MDS, a possibility that warrants further investigation.

Our finding that suppression of aberrant IRP2 activity rescued the defect in repopulation capacity of HSCs induced by FBXL5 ablation suggests that targeting of IRP2 is effective for mitigation of cellular iron overload and is therefore a candidate for therapeutic application. This notion is further supported by the finding that an increased abundance of IRP2 mRNA in CD34⁺ hematopoietic progenitor cells was related to reduced clinical survival in MDS patients without deletion of chromosome 5q. Given that complete loss of IRP2 gives rise to microcytic anaemia[42,43], however, an appropriate level of IRP2 suppression would be needed. Another limitation of such a therapeutic strategy is that an IRP2 inhibitor has not yet been developed. Given that IRP2 is an mRNA binding protein, inhibition of such binding with antisense oligonucleotides is a possible approach. Despite these limitations, inhibition of IRP2 is a potentially novel approach to the treatment of hematopoietic failure associated with FBXL5 downregulation and cellular iron overload.

## Methods

**Mice.** Generation of *Fbxl5*^F/F mice was described previously[4]. These mice were crossed with *Mx1-Cre* transgenic mice[44] or *Irp2*^−/− mice[45] to generate *Mx1-Cre/Fbxl5*^F/F and *Mx1-Cre/Fbxl5*^F/F/*Irp2*^−/− mice. All of these mice were backcrossed with C57BL/6 mice for more than six generations. Expression of Cre recombinase in mice harbouring the *Mx1-Cre* transgene was induced by intraperitoneal injection of poly(I:C) (R&D Systems, Minneapolis, MN) at a dose of 20 mg kg⁻¹ on seven alternate days beginning at 8 weeks of age. *Mx1-Cre/Fbxl5*^F/F mice and *Mx1-Cre/Fbxl5*^F/F/*Irp2*^−/− mice were analysed 4 weeks after the last poly(I:C) injection unless indicated otherwise. *Irp2*^−/− mice were analysed at 14 weeks of age. C57BL/6-Ly5.1 congenic mice were obtained from The Jackson

Laboratory (Bar Harbor, ME). For some experiments, mice were injected intraperitoneally with 5-FU (Sigma, St Louis, MO) at a dose of 150 mg kg$^{-1}$ or fed a high-iron diet formulated by supplementation of CA-1 (containing 0.03% ferric citrate; CLEA, Tokyo, Japan) with 2% ferric citrate. All mouse experiments were approved by the Animal Ethics Committee of Kyushu University.

**Flow cytometric analysis and cell sorting.** Flow cytometric analysis and cell sorting were performed with the use of FACSVerse or FACSAria instruments (BD Biosciences, San Jose, CA). Mouse antibodies to CD45.1 (A20), CD45.2 (104), Sca-1 (E13-161.7), c-Kit (2B8) or CD34 (RAM34) were obtained from BD Biosciences; those to CD3ε (145-2C11), CD4 (L3T4), CD8 (53-6.7), B220 (RA3-6B2), CD16/32 (93) or Mac1 (M1/70) were from eBioscience (San Diego, CA); and those to CD48 (HM48-1), Ter119 (TER119), Gr1 (RB6-8C5), CD71 (RI7217), CD150 (TC15-12F12.2) or Ki-67 (16A8) were from BioLegend (San Diego, CA). Antibodies to phosphorylated (Thr$^{180}$/Tyr$^{182}$) p38 MAPK (12F8) were from Cell Signaling Technology (Danvers, MA). CD4, CD8, B220, Ter119, Gr1 and Mac1 were used as lineage markers. For analysis of HSCs, antibodies except those to lineage markers were used at a 1:50 dilution and those to lineage markers were used at a 1:60 dilution. For analysis of differentiated cells, all antibodies were used at a 1:100 dilution. BM mononuclear cells flushed from the tibia and femur, thymocytes, or splenocytes of mice were suspended in phosphate-buffered saline (PBS) supplemented with 2% heat-inactivated fetal bovine serum, incubated with the indicated antibodies for 30 min on ice, washed and then analysed. For intracellular staining with antibodies to Ki-67 or to phosphorylated p38 MAPK, cells were stained for surface markers, fixed in PBS containing 2% paraformaldehyde for 20 min, permeabilized with PBS containing 0.5% saponin and 0.5% bovine serum albumin for 10 min at room temperature and then incubated with these antibodies. For detection of phosphorylated p38 MAPK, cells were further stained with Alexa Fluor 488-conjugated goat antibodies to rabbit immunoglobulin G (A11034, Molecular Probes, Eugene, OR) at a 1:100 dilution for 30 min at room temperature. For analysis of the cell cycle, cells stained with antibodies to Ki-67 were briefly exposed to Hoechst 33342 (Sigma) before analysis.

**Evaluation of intracellular iron status.** BM mononuclear cells were stained with 0.25 μM calcein-AM (Molecular Probes) in PBS for 5 min at 37 °C and then washed with PBS. The calcein-loaded cells were further stained with antibodies to surface markers for 30 min on ice, washed, incubated for 20 min on ice in PBS with or without 100 μM 2,2′-bipyridyl (Sigma), and analysed with the FACSAria instrument.

**Detection of apoptosis.** For assay of apoptosis, BM cells stained with antibodies to cell surface markers were further stained for 15 min at room temperature with annexin V and propidium iodide with the use of an Annexin V-FITC Apoptosis Detection Kit (BD Biosciences).

**Competitive reconstitution assay.** Unfractionated BM cells ($4 \times 10^5$) isolated from *Mx1-Cre/Fbxl5*$^{F/F}$, *Mx1-Cre/Fbxl5*$^{F/F}$/*Irp2*$^{-/-}$ or *Mx1-Cre/Fbxl5*$^{+/+}$ mice (CD45.2) were transplanted into lethally irradiated C57BL/6 congenic (CD45.1) recipients together with competitor BM cells ($4 \times 10^5$) from C57BL/6 congenic (CD45.1) mice. BM cells ($1 \times 10^6$) isolated from the recipient mice at 16 weeks after the first BM transfer were transplanted into a second set of lethally irradiated mice (second BM transfer).

**Homing assay.** Sorted KSL cells were incubated with 2 μM CFSE (Molecular Probes) in PBS for 12 min at 37 °C and then washed. The cells ($2 \times 10^4$) were then transplanted into lethally irradiated mice. After 16 h, BM cells were isolated from the recipient mice and analysed with the use of the FACSVerse instrument.

**Colony formation assay.** Colony formation capacity was examined for 500 KSL cells per dish with Methocult medium (MethoCult GF M3434; Stem Cell Technologies, Vancouver, BC, Canada). For a serial replating assay, cells from the first plating were collected and counted, and $1 \times 10^4$ of the cells were replated. In some experiments, FAC (100 μg ml$^{-1}$) or 10 μM DFO was added to the medium.

**RT and rtPCR analysis.** Total RNA isolated from sorted cells with the use of Isogen and Ethachinmate (Nippon Gene, Tokyo, Japan) was subjected to RT with ReverTra Ace (Toyobo, Tokyo, Japan), and the resulting cDNA was subjected to rtPCR analysis with SYBR Green PCR Master Mix and specific primers in a Step One Plus Real-Time PCR System (Applied Biosystems, Foster City, CA). Data were normalized by the abundance of β-actin mRNA (Figs 7d and 8e) or attachment region-binding protein (ARBP) mRNA (Supplementary Fig. 1a). The sequences of the various primers (sense and antisense, respectively) were as follows: 5′-AGGTGACAGCATTGCTTCTG-3′ and 5′-GGGAGACCAAAGC-CTTCATA-3′ for β-actin, 5′-GGACCCGAGAAGACCTCCTT-3′ and 5′-GCA-CATCACTCAGAATTTCAATGG-3′ for ARBP, 5′-TCTTCCTCCTGAGG-TAATGCTGTCC-3′ and 5′-CACAAAGATCCTGTTTTTGCCAGC-3′ for FBXL5, 5′-GCACCTGAGGCTGACCAATC-3′ and 5′-CATGGGCATACGGTTGTTGAG-

3′ for necdin, 5′-GCGCAAACGTCTGAGATGAGT-3′ and 5′-AGAGTTCTTC-CATCGTCCGCT-3′ for p57, 5′-CCGGACTATCAGTTGCTAA-3′ and 5′-GGACGTCGCCTGCCTGAA-3′ for HoxB5, 5′-GCTGTCCTCTAAGCGT-CACC-3′ and 5′-AGGAGCAGCAGCTCTTCTTG-3′ for MT1, 5′-CAAACC-GATCTCTCGTCGAT-3′ and 5′-AGGAGCAGCAGCTTTTCTTG-3′ for MT2, 5′-TGGGTGGAACTGCTCGTAAT-3′ and 5′-AGGATGTAGCGTCCAAATGC-3′ for Slc7a11, 5′-AAGCCGGAAATGCTGAGTTCA-3′ and 5′-GCCGTGTAGA-TATGGTACAAGGA-3′ for Hmox1 and 5′-CGGCGAGAACGAGAAGAA-3′ and 5′-AAACTTCAGACTCTTTGCTTCG-3′ for Hif1α.

**Microarray analysis.** HSCs (3,000 cells) were sorted directly into Trizol (Life Technologies, Carlsbad, CA), and total RNA was subjected to mRNA amplification, RT, fragmentation and labelling with the use of a Genechip WT Pico Kit (Affymetrics, Santa Clara, CA). Labelled single-stranded cDNA from each sample was subjected to hybridization with a GeneChip Mouse Transcriptome Array 1.0 (Affymetrics). Gene expression data were imported and analysed with the use of Transcriptome Analysis Console (TAC) Software (Affymetrics). Normalized expression data were analysed with the use of GSEA v2.0.13 software (Broad Institute, Cambridge, MA). All gene sets were obtained from the Molecular Signatures Database v4.0 distributed at the GSEA Web site (http://www.broadinstitute.org/gsea/index.jsp). Data were also analysed with IPA Software (Ingenuity Systems, Redwood City, CA).

**Haematologic and biochemical analyses.** Haematologic parameters were determined with the use of a Sysmex K-4500 automatic analyser. Serum iron concentration and total iron binding capacity were measured with a standard clinical autoanalyser. Transferrin saturation was calculated from serum iron concentration and total iron binding capacity.

**Human data analysis.** Microarray data for Lin$^-$CD34$^+$CD38$^-$CD90$^+$CD45RA$^-$ HSCs from eight MDS patients without deletion of chromosome 5q and 11 age-matched healthy control subjects were accessed at GEO with the reference series tag GSE30201 (ref. 27); those for CD34$^+$ hematopoietic progenitor cells from 183 MDS patients with various cytogenetic abnormalities and 17 healthy control subjects were accessed at GEO with the tag GSE19429 (ref. 28); and those for CD34$^+$ hematopoietic progenitor cells from MDS patients with survival data were accessed at GEO with the tag GSE58831 (ref. 29). In the latter instance, the data for 108 MDS patients without deletion of chromosome 5q and without the WHO category 'AML-MDS' who survived for >1 week were analysed. All data were downloaded for analysis of *FBXL5* (209004_s_at), *TFR1* (237214 _at), *DMT1* (1555116_s_at) or *IRP2* (214666_x_at) signal intensity.

**Analysis of published mouse data.** Microarray or RNA-sequencing data (GSE60101)[19] for FBXL5 mRNA abundance in various hematopoietic cell types are available online at Gene Expression Commons (https://gexc.riken.jp)[17] or BloodSpot (http://servers.binf.ku.dk/bloodspot)[18].

**Statistical analysis.** No statistical methods were used to predetermine sample size. Experiments were not randomized, and investigators were not blinded to allocation during experiments and outcome assessment. Quantitative data are presented as means ± s.d. as indicated and were compared between groups with the two-tailed Student's *t*-test as performed with Microsoft Excel software. Survival curves were analysed with the log-rank nonparametric test. The cutoff value to determine whether the level of *IRP2* expression was high or low in a sample (Fig. 9f) was designed by the minimal *P* value approach[46]. A *P* value of <0.05 was considered statistically significant.

**Data availability.** The microarray data were deposited in GEO under the accession number GSE93649. All other relevant data are available from the corresponding authors on reasonable request.

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

## Acknowledgements

We thank K. Miyawaki for a technical suggestion regarding microarray analysis; A. Matsumoto for a technical suggestion regarding bone marrow transplantation experiments; E. Koba, K. Tsunematsu and other laboratory members for technical assistance; and A. Ohta for help with preparation of the manuscript. This study was funded in part by KAKENHI grants (25221303 and 26640080) from the Ministry of Education, Culture, Sports, Science, and Technology (MEXT) of Japan.

## Author contributions

Y.M. planned and performed all experiments. M.N. provided materials and intellectual support. A.N. assisted with BM transplantation experiments. T.M. generated Fbxl5^F/F mice. K.I.N. coordinated the study, oversaw collection and analysis of the results, and wrote the manuscript. All authors discussed the data and commented on the manuscript.

## Additional information

**Competing interests:** The authors declare no competing financial interests.

