## [Peer Review File · Nature Communications]

Reviewers' Comments:

Reviewer #1 (Remarks to the Author)

The manuscript described the role of the cellular iron homeostasis regulator, FBXL5, in HSC maintenance. By using in vivo model, the author has demonstrated that FBXL5 depletion leads to compromised HSC's self-renewal and reconstitution capability.

The study indicated the loss of function of FBXL5 might result in hematopoietic diseases such as myelodysplasia syndrome or Fanconi anemia from expression data of patient samples. Authors also suggested iron regulatory protein 2 (IRP2) as a potential therapeutic target of the diseases, since the suppression of its accumulation in FBXL5 deficient mouse restores HSC functions.

Some specific areas to attend to are stated as below.

Major concerns:

1. How is FBXL5 expressed in different lineages of hematopoietic cells? Why the study focused mostly on Gr1+Mac1+, B220+, CD3+ lineage differentiation but ignored Ter119+ subtypes? Since the crosstalk between iron homeostasis and erythropoiesis are very well established, would FBXL5 have a more outstanding role in erythropoiesis?
2. Besides the rescue studies of repopulation capacity of FBXL5 deficient HSCs by IRP2 suppression, the potential role of IRP2 as therapeutic target would be more convincing by including MDS/FA survival data that express different levels of IRP2.

Minor concerns

1. Regarding the colony formation assay, which specific Methocult medium was used? Since different medium contains different combination of cytokines, which will lead to enrichment of different types of colonies.
2. Figure 6e shows the phospho-P38 level of control and cKO mice, besides the percentage of positive cells, what is the difference between their MFI values?
3. Given all the complications of using IRP2 as effective therapeutic target, is there any other downstream effector of FBXL5 that could also potentially mediate HSC functions?

Reviewer #2 (Remarks to the Author)

The iron-regulated F-box protein FBXL5 mediates the degradation of Iron Response Protein 2 (IRP2), thereby playing a critical role in the regulation of iron homeostasis. The authors of the present manuscript have previously published a thorough study demonstrating that the deletion of murine FBXL5 results in embryonic lethality, resulting from excessive iron accumulation. They also demonstrated that hepatic-specific ablation of FBXL5 causes mice to die when fed a high-iron diet.

In the present paper, Muto and colleagues comprehensively assessed the role of FBXL5 in the maintenance and function of hematopoietic stem cells (HSCs). Collectively, their results show that FBXL5 is required for iron homeostasis in HSCs, maintenance of HSCs, stress-induced hematopoiesis, and the repopulation capacity of HSCs. In this context, IRP2 is the major substrate of SCF-FBXL5 in HSCs (since IRP2 deletion can rescue the reconstitution of HSCs in FBXL5 deficient cells). They show that FBXL5 deficient HSCs are exposed to large amount of redox stress, suggesting that it is important to limit ROS production for maintenance of HSCs.

In all, it is a comprehensive study suggesting the importance of the cellular iron homeostasis in the maintenance of HSCs. The experiments are well controlled and the manuscript is well written. IRP2 downregulation can rescue the effect of cellular iron overload, suggesting potential for

therapeutic intervention. I could ask additional quantitations, but I believe that the authors have reached the correct conclusions. The present study sets the stage for future studies into the role of hematopoietic failure due to cellular iron overload.

Reviewer #3 (Remarks to the Author)

FBXL5 is a well-established regulator of cellular iron homeostasis. The authors and others have previously established that complete loss of FBXL5 results in severe iron accumulation, oxidative stress, and cell death. These consequences were previously shown to be largely mediated through the unregulated accumulation of the SCFFBXL5 target IRP2.

In this study, the authors have conditionally deleted FBXL5 in HSCs through crossing mice with a floxed FBXL5 allele with mice expressing Cre recombinase via the Mx1 promoter. Consistent with previous results, ablation of both FBXL5 alleles in HSCs is accompanied by indications of iron overload, impaired maintenance and self-renewal capacity. The increased cellular iron loads are likely accompanied by increased oxidative stress as indicated by enhanced oxidative stress responses (p38 MAPK phosphorylation and increased expression of oxidative stress response genes). The deleterious effects of FBXL5 deletion are largely attenuated by simultaneous deletion of IRP2. These results are entirely consistent with the known functions of FBXL5 and the sensitivity of HSCs to oxidative stress.

The primary weakness of the paper lies in the attempt to extend the clinical relevance of these findings. The authors hypothesized that FBXL5 deficiency could contribute to the development of diseases characterized by inefficient hematopoiesis. In support, FBXL5 gene expression data in HSC was presented from MDS patients, as well as published expression data from FA patients. Though differences in FBXL5 mRNA accumulation between patients versus controls met the threshold of statistical significance, the physiological significance remains highly speculative. Given that FBXL5 is highly regulated post-translationally, it's not clear that modest changes in FBXL5 mRNA levels (<10% in the case of Fig 8c) would impact functional FBXL5 activity. The potential relevance would be strengthened if the authors could demonstrate potential functional loss of FBXL5 at the level of protein or IRP2 activity (for example, correlating TfR1 mRNA levels as a potential surrogate). In addition, the authors could demonstrate that partial loss of FBXL5 is physiologically significant by assessing HSC parameters in Mx1-Cre/Fbxl5+/ Δ mice.

Response to Reviewer #1

We thank the reviewer for the careful evaluation of our manuscript and for the experimental suggestions that we feel have helped us to greatly improve our paper. Our specific responses to the points raised are as follows:

Major concerns:

1. *How is FBXL5 expressed in different lineages of hematopoietic cells? Why the study focused mostly on Gr1+Mac1+, B220+, CD3+ lineage differentiation but ignored Ter119+ subtypes? Since the crosstalk between iron homeostasis and erythropoiesis are very well established, would FBXL5 have a more outstanding role in erythropoiesis?*

[Response] To address this issue raised by the reviewer, we examined the expression of *FBXL5* in various hematopoietic cell lineages of wild-type mice by reverse transcription and real-time PCR analysis. We detected *FBXL5* mRNA in many lineages (**new Supplementary Fig. 1a**). Among differentiated cells, *FBXL5* mRNA was most abundant in myeloid (Gr1⁺Mac1⁺) cells and least abundant in the erythroid (Ter119⁺) lineage. These results are largely consistent with published data from microarray (ref. 17) (**new Supplementary Fig. 1b**) and RNA-sequencing (refs. 18, 19) (**new Supplementary Fig. 1c**) analyses.

To address the possible contribution of *FBXL5* to erythropoiesis, we examined hematologic parameters at 4 weeks (**Supplementary Fig. 2b**) or 20 weeks (**new Supplementary Fig. 3a**) after poly(I:C) injection in *Mx1-Cre/Fbx15^{F/F}* or *Mx1-Cre/Fbx15^{+/+}* mice. At both time points, parameters related to erythropoiesis (RBC count, serum hemoglobin, and MCV) did not differ significantly between *Fbx15^{Δ/Δ}* and control mice. Flow cytometric analysis also revealed no significant difference in the frequency of erythroid (Ter119⁺) cells in the bone marrow of *Fbx15^{Δ/Δ}* and control mice at 4 weeks after poly(I:C) injection (**new Supplementary Fig. 3b**).

Collectively, these results indicate that *FBXL5* has at most a minor role in erythropoiesis. In general, erythroid cells require large amounts of iron to sustain hemoglobin synthesis (ref. 41), suggesting that the importance of *FBXL5* as a brake on iron uptake might be rather limited in these cells. We have now addressed these points in the Results and Discussion sections of the revised manuscript (page 4, lines 26-33; page 6, lines 22-25; page 11, lines 18-26).

2. *Besides the rescue studies of repopulation capacity of FBXL5 deficient HSCs by IRP2 suppression, the potential role of IRP2 as therapeutic target would be more convincing by including MDS/FA survival data that express different levels of IRP2.*

[Response] We analyzed a published microarray data set for CD34⁺ hematopoietic progenitor cells (ref. 29) and found that an increased level of IRP2 mRNA was related to a poor clinical survival in MDS patients without deletion of chromosome 5q (214666_x_at) (**new Fig. 8f**). These results are therefore consistent with the notion that IRP2 is a potential therapeutic target at least for a subset of MDS patients. We have now addressed this issue in the Results and Discussion sections of the revised manuscript (page 9, line 38-page 10, line 3; page 12, lines 14-16; page 15 line 38-page 16, line 1).

Although we found that *FBXL5* expression was also down-regulated in cells of FA patients, we have now removed these data from the manuscript because they were derived from differentiated mononuclear cells and the change in expression level was modest.

Minor concerns:

1. *Regarding the colony formation assay, which specific Methocult medium was used? Since different medium contains different combination of cytokines, which will lead to enrichment*

of different types of colonies.

[Response] We apologize for the lack of this important information in the original manuscript. The colony formation assay was performed with MethoCult GF M3434 from StemCell Technologies. We have now provided this information in the Methods section of the revised manuscript (page 14, lines 8-10).

2. Figure 6e shows the phospho-P38 level of control and cKO mice, besides the percentage of positive cells, what is the difference between their MFI values?

[Response] Intracellular flow cytometric analysis revealed that the frequency of cells positive for phosphorylated p38 MAPK was greater among *Fbxl5*^{ΔΔ} HSCs than among control HSCs. The mean fluorescence intensity (MFI) of phospho-p38 for *Fbxl5*^{ΔΔ} HSCs also tended to be higher than that for control HSCs, although the difference did not achieve statistical significance ($P = 0.062$) (**new Fig. 6e**). Given that both extrinsic factors including various cytokines as well as intrinsic factors such as oxidative stress influence p38 MAPK phosphorylation status (ref. 21), we sought to examine only the influence of intrinsic factors on the level of p38 phosphorylation in HSCs. To this end, we transplanted BM cells from either *Mx1-Cre/Fbxl5*^{F/F} or *Mx1-Cre/Fbxl5*^{+/+} mice into lethally irradiated recipients and then injected these animals with poly(I:C). The frequency of *Fbxl5*^{ΔΔ} HSCs in BM of the recipients was significantly reduced compared with that of control HSCs (**new Fig. 6f**), as was the case for *Mx1-Cre/Fbxl5*^{F/F} mice treated with poly(I:C) (**Fig. 1e**). The frequency of cells positive for phosphorylated p38 MAPK as well as the MFI for phospho-p38 were significantly increased in *Fbxl5*^{ΔΔ} HSCs compared with control HSCs in the recipient mice (**new Fig. 6g**). Furthermore, flow cytometric analysis of the cell cycle revealed that the frequency of cells in the dormant state (G₀ phase) was reduced among *Fbxl5*^{ΔΔ} HSCs in the recipients compared with control HSCs (**new Fig. 7c**). These findings collectively suggest that FBXL5 deficiency increases p38 MAPK phosphorylation in as well as promotes the proliferation of HSCs in a cell-autonomous manner. We have now described these new data in the revised manuscript (page 8, lines 16-31; page 9, lines 2-4).

3. Given all the complications of using IRP2 as effective therapeutic target, is there any other downstream effector of FBXL5 that could also potentially mediate HSC functions?

[Response] FBXL5 was shown to promote degradation of CBP/p300-interacting transactivator 2 (CITED2), with depletion of FBXL5 thus resulting in an increase in the abundance of this protein (ref. 35). CITED2 interacts with the histone acetyltransferases CBP and p300 and thereby serves as a coactivator of DNA-binding transcription factors, and it promotes repression of HIF1-mediated transcription (Bhattacharya, S. *et al.*, *Genes Dev.* **13**, 64–75 (1999)). CITED2 was shown to control the proliferation of mouse embryonic fibroblasts by promoting expression of the Polycomb group genes *Bmi1* and *Mell18* (ref. 36) as well as to selectively maintain adult HSC function at least in part through regulation of p16 and p53 (ref. 37). If CITED2 accumulates in FBXL5-deficient HSCs, it might therefore serve to promote their proliferation and exhaustion. However, we believe that such a role for CITED2 is limited at most, given that the compromised stem cell capacity of *Fbxl5*^{ΔΔ} HSCs was fully restored by the additional ablation of IRP2. These observations thus suggest that IRP2 is the most promising potential therapeutic target for hematopoietic failure associated with FBXL5 down-regulation. We have now addressed this point in the revised manuscript (page 10, lines 14-30).

Response to Reviewer #2

We thank the reviewer for the positive evaluation of our manuscript.

Response to Reviewer #3

We thank the reviewer for the careful evaluation of our manuscript and for the suggestions that we feel have helped us to greatly improve our paper. Our specific responses to the points raised are as follows:

The primary weakness of the paper lies in the attempt to extend the clinical relevance of these findings. The authors hypothesized that FBXL5 deficiency could contribute to the development of diseases characterized by inefficient hematopoiesis. In support, FBXL5 gene expression data in HSC was presented from MDS patients, as well as published expression data from FA patients. Though differences in FBXL5 mRNA accumulation between patients versus controls met the threshold of statistical significance, the physiological significance remains highly speculative. Given that FBXL5 is highly regulated post-translationally, it's not clear that modest changes in FBXL5 mRNA levels (<10% in the case of Fig 8c) would impact functional FBXL5 activity. The potential relevance would be strengthened if the authors could demonstrate potential functional loss of FBXL5 at the level of protein or IRP2 activity (for example, correlating TFR1 mRNA levels as a potential surrogate). In addition, the authors could demonstrate that partial loss of FBXL5 is physiologically significant by assessing HSC parameters in Mx1-Cre/Fbxl5+/ Δ mice.

[Response] We would like to clarify that “Relative signal intensity” for the published human microarray analyses (**Fig. 8a–e**) is not linearly correlated with the abundance of the corresponding mRNA. For example, although the relative *FBXL5* signal intensity appears to be reduced by only ~25% in Figure 8a (GSE30201), analysis of the data with Transcriptome Analysis Console (TAC) Software (Affymetrics) revealed that the estimated difference in the amount of *FBXL5* mRNA between the MDS and control HSCs was 13.77-fold ($P < 0.001$). These results thus suggest that the abundance of *FBXL5* mRNA is substantially down-regulated in HSCs of a subset of MDS patients. The partial loss (probably ~50%) of *FBXL5* in *Fbxl5*^{+/-} mice was found to have a minimal effect on the abundance of *FBXL5* target proteins (ref. 4). We therefore assume that an ~50% reduction in *FBXL5* expression (not relative *FBXL5* signal intensity) probably does not affect the pathogenesis of MDS.

As suggested by the reviewer, we examined the expression levels of *TFR1* (237214_s_at) and *DMT1* (1555116_s_at) as potential surrogates of IRP2 activity. We found that the relative signal intensities for these genes in published microarray data (GSE30201) were significantly ($P < 0.01$) increased for MDS patients (**new Fig. 8b, c**), suggesting that IRP2 activity is indeed elevated in MDS HSCs compared with control HSCs. The *TFR1* signal intensity was also significantly ($P < 0.05$) increased in RARS patients (**new Fig. 8e**). We have now addressed these points in the revised manuscript (page 9, lines 23-26, 32-33).

Although we found that *FBXL5* expression was also down-regulated in cells of FA patients, we have now removed these data from the manuscript because they were derived from differentiated mononuclear cells and the change in expression level was modest.

Reviewers' Comments:

Reviewer #1:

Remarks to the Author:

After reading through their responses to our major and minor concerns, I think their response is concise and accurate. They are able to refer to literature to substantiate their responses, as well as to point out where the revision is in the manuscript precisely.

Reviewer #3:

Remarks to the Author:

The manuscript has been improved by the addition of the TFR1 and DMT1 data, and the removal of the FA data, in Fig. 8.